# ASMR: Angular Support for Malfunctioning Client Resilience in Federated Learning

**Mirko Konstantin**[1]                    MIRKO.KONSTANTIN@GRIS.TU-DARMSTADT.DE

[1] *Technical Univerity Darmstadt*

**Moritz Fuchs**[1]                    MORITZ.FUCHS@GRIS.TU-DARMSTADT.DE

**Anirban Mukhopadhyay**[1]                    ANIRBAN.MUKHOPADHYAY@GRIS.TU-DARMSTADT.DE

**Editors:** Accepted for publication at MIDL 2024

## Abstract

Federated Learning (FL) allows the training of deep neural networks in a distributed and privacy-preserving manner. However, this concept suffers from malfunctioning updates sent by the attending clients that cause global model performance degradation. Reasons for this malfunctioning might be technical issues, disadvantageous training data, or malicious attacks. Most of the current defense mechanisms are meant to require impractical prerequisites like knowledge about the number of malfunctioning updates, which makes them unsuitable for real-world applications. To counteract these problems, we introduce a novel method called Angular Support for Malfunctioning Client Resilience (ASMR), that dynamically excludes malfunctioning clients based on their angular distance. Our novel method does not require any hyperparameters or knowledge about the number of malfunctioning clients. Our experiments showcase the detection capabilities of ASMR in an image classification task on a histopathological dataset, while also presenting findings on the significance of dynamically adapting decision boundaries.

**Keywords:** Federated Learning, Outlier Detection

## 1. Introduction

Federated Learning (FL) has become an emerging research topic in the last few years (Mammen, 2021). Due to the concept of training models in a distributed manner, FL comes with a bunch of applications in various fields (Yang et al., 2019), such as medical imaging (Rieke et al., 2020). Apart from dealing with data regulations (Truong et al., 2021), access to annotated and heterogeneous data is a major challenge in medical imaging (Willemink et al., 2020), as model robustness depends on it. FL addresses the challenge of extending training data across multiple institutions (Guo et al., 2021), mitigating the need for extensive annotations per institution and thereby reducing the associated costs (Tajbakhsh et al., 2021).

However, there is **no guarantee about the utility of these updates** (Ma et al., 2021a). Local models trained under unfavorable conditions can negatively affect the aggregated global model (Wagner et al., 2022). Updates leading to a degradation in global model performance are termed malfunctioning and can be categorized into two distinct categories, as demonstrated in Figure 1. As demonstrated by (Jere et al., 2021), clients may exhibit **malicious** behavior to intentionally corrupt the global model performance by tampering with the updates before transmission. Furthermore, **unreliable** clients, as indicated by (Foucart et al., 2018), may unintentionally send malfunctioning updates, facing challenges

like broken devices, transmission errors, or faulty image acquisition (Kanwal et al., 2022). The unpredictable nature of these issues renders knowledge about the precise number of malfunctioning updates impractical. Corruptions do not necessarily mean that the global model diverges immediately, but have a significant impact on model performance in the long term. This makes malfunctioning updates hard to detect without knowledge about the baseline performance (Shejwalkar et al., 2022). To overcome this challenge, an algorithmic solution is required. In the past, several works have been published to detect and exclude malfunctioning updates from aggregation (Blanchard et al., 2017) (Shejwalkar and Houmansadr, 2021) (Sattler et al., 2020) (Li et al., 2023). These approaches frequently entail **impractical prerequisites**. Certain methodologies necessitate access to a publicly available dataset (Li et al., 2020) for generating reference updates used in training a classification model designed to identify malfunctioning updates. Conversely, others *require knowledge about the constant number of malfunctioning updates* per round (Blanchard et al., 2017; Shejwalkar and Houmansadr, 2021). Recognizing these research gaps, we introduce the novel concept of **angular client support**. This concept allows us to propose **an out-of-the-box solution** called Angular Support for Malfunctioning Client Resilience (ASMR) that can reliably detect malfunctioning updates. The **number of excluded updates is adapted dynamically** each round, **without required knowledge** about the population of malfunctioning clients. Furthermore, knowledge about the test data or the evaluation protocol is not required. The principle of **angular client support** introduces a novel perspective emphasizing the interconnectedness between clients based on their nature.

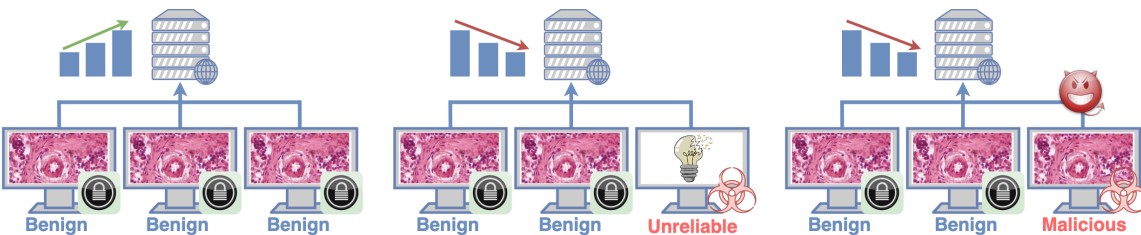

Figure 1: The leftmost FL system exemplifies an ideal scenario, showcasing optimal performance. In the middle system, an unreliable client grapples with technical issues, resulting in the degradation of the global model performance due to its updates. Meanwhile, the system on the right features a malicious client intentionally corrupting its updates.

This concept underscores the importance of fostering collaborative relationships among benign clients to enhance collective resilience against malfunctioning counterparts. The closer clients are in angular distance, the more robust their collaborative support becomes. In this work, we assume the malfunctioning updates to be independent. In particular, *malicious clients do not exchange knowledge* with each other to improve the effectiveness of their attacks. Furthermore, we assume that malicious clients do not have knowledge about the other clients in terms of a number of malicious, unreliable, or benign clients. This indicates that even a majority of malfunctioning clients do not mean, that malfunctioning updates support each other. In this case, a **supportive minority of benign updates**

enables ASMR to **detect malfunctioning updates** and maintain a **steady convergence** during training. To demonstrate the detection capabilities of ASMR, we applied it to three cases. First, we consider the case of **malicious clients**. We selected a subset of clients that perform untargeted attacks, such as **Additive Noise Attacks (ANA)** and **Sign Flipping Attacks (SFA)**. Second, we apply ASMR to the case of **unreliable clients**. The selected subset of clients train their local model on data, for which we simulate device failure and acquisition errors by augmenting the data with pathology-specific artifacts, that significantly degrade the accuracy. Ultimately, the combination of both previous cases is considered. The selected group of clients is either a malicious or an unreliable client. Our novel approach is compared to three other state-of-the-art (SOTA) detection algorithms on an image classification task in the field of digital pathology. **The code is available at:** https://github.com/MECLabTUDA/ASMR.

## 2. Background

Federated Learning became a serious asset in medical imaging in recent years (Rieke et al., 2020) (Ng et al., 2021) (Nguyen et al., 2022). Locally trained models are sent to a central server and aggregated with algorithms like FedAvg (Khan et al., 2021) (Ye et al., 2020) or FedAvgM (Hsu et al., 2019). Federated learning systems are vulnerable to malicious clients that perform attacks to corrupt the global model performance though.

**Malicious Client Detection:** To overcome this vulnerability against the aforementioned attacks, defense mechanisms were introduced. In 2017 (Blanchard et al., 2017) proposed Multi-Krum (MKrum). MKrum chooses the updates that minimize the squared distance to its nearest neighbors. However, this technique requires an estimated number of malicious clients. Later, (Shejwalkar and Houmansadr, 2021) proposed Divide-and-Conquer (DnC), which deals with determining principal components and computing the projections of the updates. Afterward, the updates with the largest projections are excluded from aggregation. Like MKrum, DnC requires a number of malicious clients. Meanwhile, (Sattler et al., 2020) implemented a clustering approach known as CFL, aimed at uncovering hidden clustering among clients in FL systems to differentiate between benign and malicious clients. Due to its inherent clustering mechanism, CFL does not necessitate prior knowledge about the number of malicious clients. Likewise,(Li et al., 2021) utilize k-means clustering for the detection of malfunctioning clients, which, however, exhibits less resilience to noise and is incapable of effectively handling clusters of varying sizes according to (Sisodia et al., 2012). Approaches like ShieldFL (Ma et al., 2022) or SFAP (Ma et al., 2021b) rely on access to training data, stored by the server. Nonetheless, in the domain of medical imaging, where centralized data storage is often unfeasible. FLTrust, as proposed by (Cao et al., 2020), operates under the assumption of a trusted round phase devoid of malfunctioning clients. This assumption may not always hold, e.g. if certain clients have inherently flawed data acquisition processes from the beginning.

**Histopathology:** Computational Histopathology has the promise of elevating the workload from pathologists and accelerating the process of delivering accurate diagnosis and prognosis to patients (Couture et al., 2018; Griem et al., 2023). For this process to be applicable, tissues require tissue fixation, processing, cutting, staining, and digitization, which are subject to many different kinds of artifacts (Kanwal et al., 2022). Even though

the heterogeneity in tumor tissues can still be limiting, the heterogeneity intensifies across multiple institutions, thereby underscoring the value of federated learning (Wagner et al., 2022). Respectively, a centralized system, holding just the local data, might not generalize and be robust to different stainings and artifacts (Faryna et al., 2021), even when they are synthetic (Babendererde et al., 2023), as these cause *silent failures* in many models.

## 3. Methodology

In this paper, we introduce the concept of **angular client support** for malfunctioning client detection in federated learning systems. Based on this concept, a new detection technique, called ASMR, is proposed that aims to protect the system against malfunctioning updates. Moreover, ASMR establishes a dynamic decision boundary, selectively excluding updates from aggregation, thereby eliminating the requirement for additional hyperparameters. This leads to automatic adaption in terms of a changing number of malicious clients.

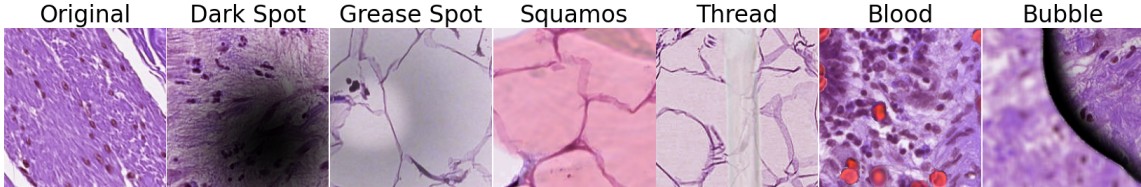

Figure 2: The first image shows an image without artifacts while the other demonstrates the artifacts that are used in this work.

**Malfunctioning Updates:** Malicious clients, that aim to degrade the global model performance intentionally, may send corrupt their updates before sending them to the server. Such kinds of attacks are called untargeted attacks, which can go undetected for a long time (Shejwalkar et al., 2022). To achieve the attackers' goal, they may change the labels of their local training data to make the model inaccurate (Bhagoji et al., 2019), or they send a random update, which is not aligned to the task at all (Fang et al., 2020). The two common baseline attacks that are considered in this work are the ANA (Li et al., 2019) (Wu et al., 2020) and SFA (Li et al., 2019) (Wu et al., 2020). In our implementation, for the ANA, we introduced noise resulting in a global model performance decrease ranging from 20% to 30%. For the SFA, we multiplied the update by a negative constant to invert the direction of the gradients. The incorporation of even a minimal number of SFAs has the potential to induce divergence in the global model. This renders SFAs unforgiving towards uncertainties in detection, yet they are anticipated to be more easily detected. The third type of malfunctioning we consider is pathology-specific artifacts, which we simulate by the FrOoDo (Stieber et al., 2022) framework adding artifacts to local training data. Figure 2 demonstrates the selected artifacts, e.g. blood cells that cover the tissues or grease spots on the probes. The artifacts were chosen such that the global model performance is affected by 50% - 60%. A table has been included in Appendix B, outlining the details of the malfunctioning updates to provide a comprehensive overview.

**Angular Client Support:** The notion of angular client support describes how clients are connected based on their angles, as depicted in Figure 3. The close angular proximity

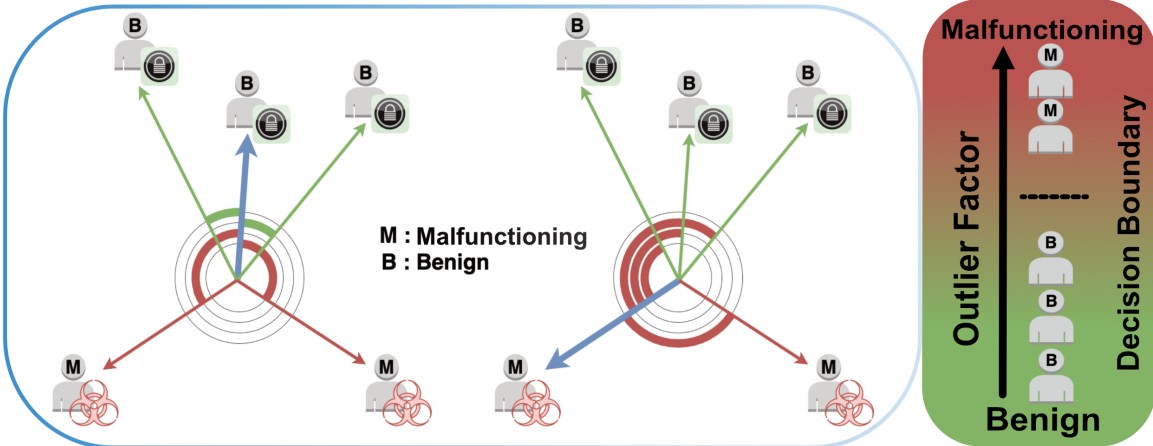

Figure 3: This figure demonstrates the idea behind client support. The clients labeled with M hold malfunctioning updates, while the others denoted with B hold benign updates. The angles between the update, denoted with the blue arrow, to all other updates are visualized. The green angles are the supporting ones.

among benign clients signifies mutual support, contrasting with malfunctioning clients that lack such support. Additionally, it is noteworthy that smaller angles between benign updates indicate stronger support among them. This concept draws inspiration from the research of (Geiping et al., 2020), who asserted that the angle between gradients conveys information about the prediction change. Hence, we posit that the angular distance of a malfunctioning update must be significantly distant from benign ones in order to exert a detrimental impact on the global model performance.

**ASMR:** This method identifies malfunctioning clients based on their **local model parameters**. Initially, all received updates undergo normalization by dividing the vectors by their magnitude. Subsequently, the pairwise cosine distance ($cosDist$) between local model parameters is computed. Then the outlier factor ($OF$) is determined for each update, by calculating it inspired by (Breunig et al., 2000). The reachability density ($rd(\bullet)$ in Eq 1) is computed for each update, representing the inverse of the average $cosDist$, and taking into consideration **all** updates.

$$rd(p) = 1/(\frac{\sum_{o \in N(p)} cosDist(p,o)}{|N(p)|}) \qquad (1)$$

where $N(p)$ is the set of all clients except $p$, and $|\bullet|$ defines the cardinality of the set $\bullet$. Using the reachability density ($rd(\bullet)$), the outlier factor is determined by:

$$OF(p) = \frac{\sum_{o \in N(p)} \frac{rd(o)}{rd(p)}}{|N(p)|} \qquad (2)$$

Subsequently, the updates undergo ordering based on their outlier factors, following which the decision boundary is established as the most substantial gap between two suc-

cessive updates. The subset of updates exhibiting higher outlier factors is consequently identified as malfunctioning and is thereby excluded from the pool. Finally, any aggregation algorithm of choice can be applied to the set of benign updates. This characteristic has the potential to hold even in scenarios of a majority of malfunctioning clients that support each other less than the benign ones. As long as benign clients maintain superior support, ASMR is anticipated to exhibit robust protective capabilities, encompassing precise detection and automated adjustment to the prevalence of malfunctioning updates.

## 4. Experiments

In this section, we commence by presenting the dataset, metrics, and training particulars employed in this study. Subsequently, we conduct a comprehensive evaluation of ASMR, juxtaposed with three alternative methods.

**Dataset:** We used one dataset of the histopathology domain to evaluate our approach. The colorectal cancer dataset (CRC) (Kather et al., 2018) contains 100,000 images with a resolution of $224 \times 224$ extracted from 86 human cancer tissue slides. The corresponding classification task covers nine different classes. Those are adipose(ADI), background (BACK), debris (DEB), ymphocytes (LYM), mucus (MUC), smooth muscle (MUS), normal colon mucosa (NORM), cancer-associated stroma (STR), colorectal adenocarcinoma epithelicum (TUM). We applied a random 0.7 / 0.3 train test split. Each client gets an equally sized portion of the training data. For this classification task, a Resnet50 (He et al., 2016) architecture was chosen, with pretrained weights from ImageNet (Deng et al., 2009).

**Evaluation:** In our analysis, the False Positive Rate (FPR) and True Positive Rate (TPR) were utilized to gauge the efficacy of malfunctioning update detection. Ultimately, we analyze the final test accuracy of the global model after twelve training rounds. The metrics are assessed over ten seeds for experiments with a fixed number of clients and five seeds for scenarios involving a dynamically changing number of clients.

**Training details:** In this work, we consider a scenario of ten clients, where a subset of clients is selected to send malfunctioning updates. In our experimental setup, clients transmit their local model parameters as updates. The server utilizes FedAvg for aggregation to derive the global model. Prior to sending an update to the server, each client undergoes training for one local epoch. To illustrate the serious adverse effects of three malfunctioning clients, we put a figure in the Appendix C. It is expected that the severity of the malfunctioning updates correlates with the difficulty of detecting them. Therefore, SFAs are expected to be easier to detect than ANAs. To prevent the global model from negative effects, detection algorithms are applied to the system. ASMR is compared to MKrum (Blanchard et al., 2017), DnC (Shejwalkar and Houmansadr, 2021), and CFL (Sattler et al., 2020) to demonstrate the effectiveness. As mentioned earlier, DnC and MKrum require a parameter that specifies how many updates should be excluded from each round. For this scenario, we set the parameter to three, such that MKrum and DnC match their optimal prerequisites. Consequently, we establish a predetermined count of three malicious clients for our evaluations and set the number of excluded clients in MKrum and DnC to three. To underscore the significance of a dynamically adapting decision boundary, we explored one more scenario involving a variable count of malfunctioning clients. Specifically, we set the malfunctioning clients to four, each sending a malfunctioning update with a 75%

probability. Notably, MKrum and DnC persistently exclude three clients per round, maintaining prior knowledge of the expected number of malfunctioning clients. We aim to show that ASMR is robust against this scenario. We put a table with these details in Appendix A. The experimental framework is developed using PyTorch (Paszke et al., 2019), and all experiments were carried out on *Nvidia A100* GPUs.

## 4.1. Results

In our experiments, our objective is to demonstrate that ASMR attains comparable results, even if MKrum and DnC operate in their comfort zone, relying on the **impractical** knowledge about the number of malfunctioning clients. Additionally, our goal is to demonstrate that ASMR surpasses CFL, a method that also autonomously establishes the decision boundary.

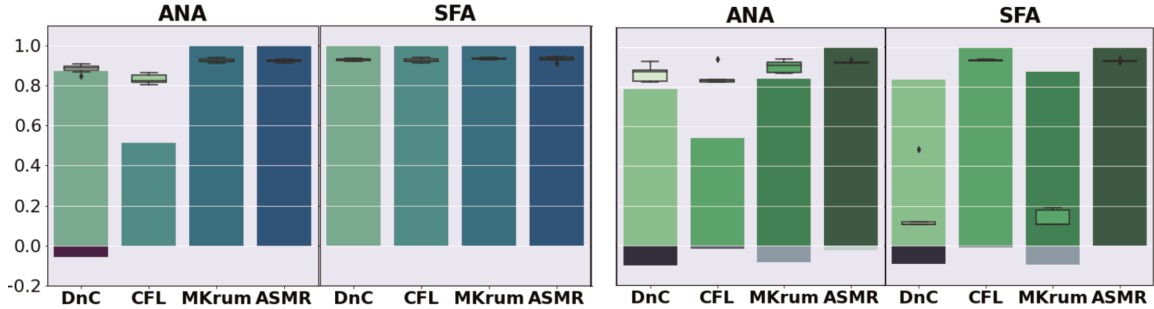

Figure 4: The left plots depict a fixed number of malicious updates, while the right ones illustrate a dynamically changing count. Bar plots represent TPRs in the positive direction and FPRs in the negative. The distribution of the final global model performance is shown through a boxplot.

**Malicious Clients - Untargeted Attacks:** This experiment investigates malicious clients in the context of two distinct attacks: ANA and SFA. Each attack is examined independently, and the outcomes are visually presented in Figure 4. We present comprehensive results in a table in the Appendix C. ASMR demonstrates superior performance in detecting ANA compared to CFL and DnC, even when the number of ANA per round is fixed. Notably, our method exhibits robustness across both scenarios, seamlessly adapting to dynamic changes in the number of malicious clients without performance degradation.

**Unreliable Clients - Pathology Specific Data Artifacts:** In the second scenario, we analyze the impact of unreliable clients that train on data containing artifacts. The results are visualized in Figure 5 and comprehensive results are provided in a table in the Appendix C. Notably, only our method and CFL exhibit an effective detection rate in both scenarios. The absence of detection of MKrum and DnC in the case of a dynamically changing number of unreliable clients significantly impairs the global model performance.

**Malfunctioning Clients - Untargeted Attacks combined with Artifacts:** In the final case, we assessed the general scenario of malfunctioning clients, encompassing those employing an ANA, SFA, or training on data with artifacts. The results are pre-

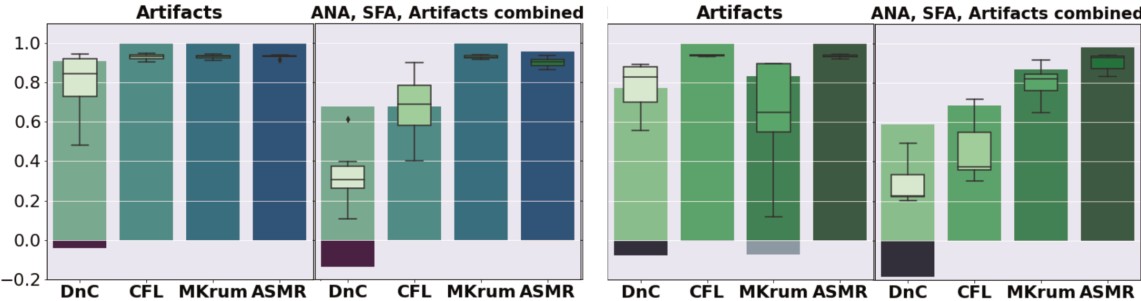

Figure 5: The left plots depict a fixed number of malfunctioning updates, while the right ones illustrate a dynamically changing count. Bar plots represent TPRs in the positive direction and FPRs in the negative. The distribution of the final global model performance is shown through a boxplot.

sented in Figure 5 and comprehensive results are in Appendix C. In scenarios with full but impractical knowledge about the number of malfunctioning clients, MKrum demonstrates slightly superior performance, leading to an average accuracy difference of 2.8% compared to ASMR. However, ASMR is the only method that exhibits robustness against scenarios with a dynamically changing number of malfunctioning clients.

## 5. Conclusion

In this work, we systematically explore various instances of malfunctioning updates that can compromise the integrity of the global aggregated model within a federated learning system. Malicious clients may intentionally degrade the global model, while unreliable clients may train on disadvantageous data. Our findings underscore the deleterious impact of incorporating malfunctioning clients into the aggregation process. To mitigate these negative effects on the global model, we propose an out-of-the-box solution named ASMR that circumvents the need for hyperparameters or prerequisites, while setting an automatically adapting decision boundary for excluding clients from aggregation. This method effectively detects and excludes malfunctioning updates from the aggregation process. Our results demonstrate that our approach offers protection capabilities comparable to or better than state-of-the-art methods, even when those methods rely on unrealistic but necessary knowledge about the number of malfunctioning clients. We additionally presented findings highlighting the significance of an automatically adapting decision boundary, demonstrating the robustness of our method in the face of a dynamically changing number of malfunctioning updates. In summary, our experiments were conducted on a homogenous dataset featuring significant cases relevant to medical imaging. Moving forward, it is imperative to explore additional cases, particularly those involving heterogeneous datasets.

## Acknowledgments

This work was supported by the Bundesministerium für Bildung und Forschung (BMBF) with grant [01KD2210B].

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

## Appendix A. Training Details

We put two tables here to give a structured overview about the classification of malfunctioning updates 1 and the details of our training setup 2

## Appendix B. Impact of Malfunctioning Clients

Figure 6 demonstrates the impact if no defense mechanism is applied to a FL system containing three malfunctioning clients.

Table 1: Malfunctioning Clients

| Malfunction | Type | Severity | Global Model Decrease | Implementation |
|---|---|---|---|---|
| Malicious | SFA | Strong | random performance | Changes direction of vectors |
| Malicious | ANA | Low | 20-30% | Adds Gaussian noise |
| Unreliable | Artifacts | Middle | 50-60% | Artifacts on training data |

Table 2: Experimental Setup

| | Fixed | Dynamic |
|---|---|---|
| Total clients | 10 | 10 |
| Malfunctioning clients | 3 | 4 |
| Malfunctioning probability | 100% | 75% |
| Local epochs | 1 | 1 |
| Aggregation method | FedAvg | FedAvg |
| Data split | random 0.7 / 0.3 train test split | |

## Appendix C. Results

The subsequent tables present exhaustive results from the conducted experiments. In the notation, *Fixed* signifies experiments involving a consistent number of malfunctioning clients, while *Dynamic* indicates experiments where the number of malfunctioning clients varies dynamically. The reported values represent averages across the assessed seeds.

Table 3: Malicious Clients

| | Fixed | | | | | | Dynamic | | | | | |
|---|---|---|---|---|---|---|---|---|---|---|---|---|
| | ANA | | | SFA | | | ANA | | | SFA | | |
| Methods | TPR | FPR | Acc | TPR | FPR | Acc | TPR | FPR | Acc | TPR | FPR | Acc |
| DnC | .872 | .055 | .885 | 1. | .0 | .928 | .789 | .099 | .868 | .835 | .091 | .185 |
| CFL | .514 | .0 | .832 | 1. | .0 | .926 | .542 | .019 | .852 | 1. | .007 | .934 |
| MKrum | 1. | .0 | .926 | 1. | .0 | .933 | .842 | .084 | .902 | .877 | .096 | .138 |
| ASMR. | 1. | .001 | .924 | 1. | .0 | .932 | 1. | .024 | .925 | 1. | .0 | .931 |

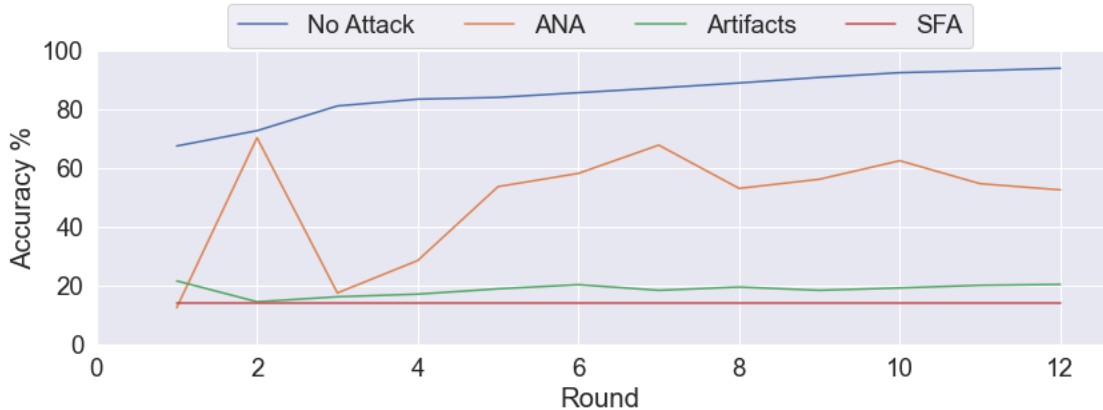

Figure 6: These graphs showcase the deleterious effects on the test accuracy of the global model over twelve rounds, underscoring the impact of three (out of ten) malfunctioning clients, operating without safeguards.

Table 4: Unreliable and Malfunctining Clients

| | Fixed | | | | | | Dynamic | | | | | |
|---|---|---|---|---|---|---|---|---|---|---|---|---|
| | Artifacts | | | Combined | | | Artifacts | | | Combined | | |
| Methods | TPR | FPR | Acc | TPR | FPR | Acc | TPR | FPR | Acc | TPR | FPR | Acc |
| DnC | .908 | .039 | .8 | .678 | .14 | .321 | .771 | .078 | .772 | .59 | .184 | .296 |
| CFL | 1. | .0 | .929 | .675 | .0 | .674 | 1. | .005 | .937 | .685 | .039 | .46 |
| MKrum | 1. | .0 | .931 | 1. | .0 | .928 | .833 | .071 | .621 | .868 | .095 | .798 |
| ASMR. | 1. | .0 | .931 | .956 | .006 | .9 | 1. | .0 | .934 | .98 | .04 | .901 |

