# OpenReview forum: "ASMR: Angular Support for Malfunctioning Client Resilience in Federated Learning"
_MIDL.io/2024/Conference — MIDL 2024 Poster_

### Official Review · Reviewer_EW71 · 2024-02-27

**Confidence:** 3
**Preliminary Rating:** 3
**Final Rating:** 3.5

**Summary:**

The paper aims to address the challenge of malfunctioning updates in Federated Learning (FL), which degrade the global model's performance due to technical issues, poor data quality, or malicious attacks. Traditional defence mechanisms often require impractical prerequisites, such as knowledge of the number of malfunctioning updates, making them unsuitable for real-world applications. The authors propose a novel method named Angular Support for Malfunctioning Client Resilience (ASMR) that dynamically excludes malfunctioning clients based on their angular distance, eliminating the need for heavy hyper-parameters tunning or prior knowledge about the number of malfunctioning clients.

Experiments conducted on an image classification task using a histopathological dataset demonstrate ASMR's capability to detect malfunctioning updates and maintain model performance by adapting decision boundaries dynamically. In experiments, the authors demonstrate that the proposed ASMR offers a more flexible and effective solution compared to previous methods.

**Strengths:**

- Significant and Clear Motivation: The paper addresses the crucial challenge of detecting malicious clients in Federated Learning (FL), especially in medical imaging, by introducing a simple, angular distance-based method. This approach significantly reduces the need for complex prior knowledge, highlighting its practical relevance and impact.
- Simple and Effective Methodology: The proposed method is straightforward and is easy to understand. The proposed method is demonstrated to be effective across various attack scenarios.

**Weaknesses:**

- The authors seem to lack some key descriptions regarding the division of clients in the experiments. The heterogeneity differences between the clients' data should have a significant impact on the method, as it affects the angular direction of the client updates.
- An expanded discussion or consideration of other similarity-based algorithms for detecting malicious models is warranted. This should include other statistical metrics such as Euclidean distance [1], cosine similarity [2], k-means clustering [3], and the Pearson correlation coefficient [4].

[1] Cao, Xiaoyu, et al. "Fltrust: Byzantine-robust federated learning via trust bootstrapping." arXiv preprint arXiv:2012.13995 (2020).
[2] Ma, Zhuoran, et al. "ShieldFL: Mitigating model poisoning attacks in privacy-preserving federated learning." IEEE Transactions on Information Forensics and Security 17 (2022): 1639-1654.
[3] Li, Dongcheng, et al. "Detection and mitigation of label-flipping attacks in federated learning systems with KPCA and K-means." 2021 8th International Conference on Dependable Systems and Their Applications (DSA). IEEE, 2021.
[4] Ma, Zhuoran, et al. "Pocket diagnosis: Secure federated learning against poisoning attack in the cloud." IEEE Transactions on Services Computing 15.6 (2021): 3429-3442.

**Detailed Comments:**

The presentation of Figure 4 could be further enhanced by adjusting the axes to reduce unnecessary white space on the sides, and by adding a necessary introduction to the key methods in the caption, indicating which method needs to be highlighted.

**Justification Of Final Rating:**

I am satisfied with the author's discussion about concerns in the weakness. It would be better if this part could be included in the paper, and it would be even better if the author could provide relevant experimental data to support their claims.

**Justification Of The Preliminary Rating:**

My main concern is that the effectiveness of this method may be limited to specific situations of data heterogeneity (for example, it may only work when data heterogeneity is small). Also, the comparison in the experiments with other similarity based method is not comprehensive enough.

**Questions To Address In The Rebuttal:**

- Can you explain in detail why the performance of the ANA curve in Figure 4 is so good in the 2nd round that it surpasses the performance of the subsequent rounds?

---

> ### Author Response · Authors · 2024-03-17
>
> ### The authors seem to lack some key descriptions regarding the division of clients in the experiments. The heterogeneity differences between the clients' data should have a significant impact on the method, as it affects the angular direction of the client updates.
>
> Thanks for your suggestion, we revised the manuscript and added details for the training in Table 2. It is true that a distribution shift between clients affects the angulars between local model parameters. Inspired by Li et al.[1], we argue that, when a malfunctioning update is able to perform serious damage on the global model, by disturbing the average in FedAvg. Then this update must be far away from all other updates. The shift of a malfunctioning client needs to be way larger than a distribution shift between clients, in order to have a performance degrading impact.  “Malfunctioning updates” that do not cause serious damage to the global model do not need to be defended, as they no longer fit the definition of being malfunctioning and a threat to the global model.
> [1] Li, Suyi, et al. "Learning to detect malicious clients for robust federated learning." arXiv preprint arXiv:2002.00211 (2020).
>
> ## Detailed Comments
>
> ### The presentation of Figure 4 could be further enhanced by adjusting the axes to reduce unnecessary white space on the sides, and by adding a necessary introduction to the key methods in the caption, indicating which method needs to be highlighted.
>
> Thank you for your suggestions, we updated the figure accordingly and due to the limitation of space, we moved Figure 4 (now Figure 6) )to the appendix.
>
> ## Questions To Adress In The Rebuttal
>
> ### Can you explain in detail why the performance of the ANA curve in Figure 4 is so good in the 2nd round that it surpasses the performance of the subsequent rounds?
>
> Figure 4 demonstrate the impact of three malfunctioning clients that are included in the aggregation each round (no detection algorithm applied).The ANA adds noise to the model parameters. Especially in an earlier phase of the training where the global model is not converged yet, additional noise to the model parameters might have a positive impact on the performance. As it has been shown that (weak) noise regularization can have a positive impact on model training and robustness [7]. However, over time of the training in a federated learning, it rather leads to instability of the training than decrease the performance consistently, as the noise is stronger due to the natural diversity of clients. That is why in the second round it is for example very close to the no attack scenario, while the performance of round three is almost at random performance. Therefore there is no consistency in the training leading to severely worse performance in the long run.
> [7] Noh, Hyeonwoo, et al. "Regularizing deep neural networks by noise: Its interpretation and optimization." Advances in neural information processing systems 30 (2017).

---

> ### Author Response · Authors · 2024-03-17
>
> ### An expanded discussion or consideration of other similarity-based algorithms for detecting malicious models is warranted. This should include other statistical metrics such as Euclidean distance [1], cosine similarity [2], k-means clustering [3], and the Pearson correlation coefficient [4].
>
> Thank you for this valuable input. We discussed these methods in depth here and added part of it to the background section to reflect on these methods.
>
> - **FLTrust**:
> Uses the Euclidean distance as a measure. We decided on the cosine distance in this context because it tends to provide more meaningful comparisons in the high dimensional space, as shown by [5].
> In this work, a trust score is computed that weights the updates during aggregation. This means malfunctioning clients are not really detected, but it is aimed that the impact of such clients is reduced. It leaves the potential that malfunctioning clients slowly but progressively worsen the corruption of the global model with each consecutive round.The trust score is based on a reference model trained by the server. For Federated Learning protocol in medical images it is not always feasible to have a public dataset that aligns with the task at hand to be present, as annotation costs can be notoriously high. Therefore, we do not assume any knowledge of training data by the server in our work.
>
>  - **ShieldFL**
> This approach is an approach which is robust against non-iid data. It uses the cosine similarity as a measure, similar as our method.
> However, this is again not a detection algorithm but an approach to reduce the negative impact of malfunctioning clients during aggregation, leaving the potential for degrading the global model under a majority of malfunctioning clients.
> In this approach, the updates are compared with the central server model. They state that this approach requires at least one trusted training round in the absence of malfunctioning clients. We cannot see that this can be guaranteed. It is a strong requirement, which is needed to guarantee any kind of security.
>
>
> - **K-means & KPCA**
> This methodology employs k-means clustering to differentiate between benign and malfunctioning clients. [6] delineates the limitations of k-means, notably its susceptibility to noise and its incapacity to handle clusters of varying sizes, among other factors.These limitations are particularly significant in federated learning, especially when considering malfunctioning clients. Malfunctioning clients, though in the minority, introduce noise and can result in clusters with varying sizes.
>
>
> - **SFAP**
> This approach uses a loss function to select the benign clients for aggregation. The loss is calculated over an encrypted validation set. The decision boundary is set as a hyperparameter.
> The absence of malfunctioning clients is assumed, when collecting the dataset. We do not see that this can be guaranteed, e.g. clients with faulty data aquisition. Unlucky data acquisition during the validation set collection phase would break this defense approach.
> Consequently, the choice of a loss function for a score calculation might not the best choice, since it can not be guaranteed that there is access to such a dataset. Furthermore, the choice of the decision boundary is difficult and impractical, as an unbiased dataset would be required to evaluate it, leading to separate hold-out set from the test set in order to not pose the risk of data leakage.. Even though there is a recommendation for a boundary, this may change from dataset to dataset, especially in the case of non-iid data.
>
> [1] Cao, Xiaoyu, et al. "Fltrust: Byzantine-robust federated learning via trust bootstrapping." arXiv preprint arXiv:2012.13995 (2020).
>
> [2] Ma, Zhuoran, et al. "ShieldFL: Mitigating model poisoning attacks in privacy-preserving federated learning." IEEE Transactions on Information Forensics and Security 17 (2022): 1639-1654.
>
> [3] Li, Dongcheng, et al. "Detection and mitigation of label-flipping attacks in federated learning systems with KPCA and K-means." 2021 8th International Conference on Dependable Systems and Their Applications (DSA). IEEE, 2021.
>
> [4] Ma, Zhuoran, et al. "Pocket diagnosis: Secure federated learning against poisoning attack in the cloud." IEEE Transactions on Services Computing 15.6 (2021): 3429-3442.
>
> [5] Shirkhorshidi, Ali Seyed, Saeed Aghabozorgi, and Teh Ying Wah. "A comparison study on similarity and dissimilarity measures in clustering continuous data." PloS one 10.12 (2015): e0144059.
>
> [6] Sisodia, Deepti, et al. "Clustering techniques: a brief survey of different clustering algorithms." International Journal of Latest Trends in Engineering and Technology (IJLTET) 1.3 (2012): 82-87.

---

### Official Review · Reviewer_1Cnm · 2024-03-01

**Confidence:** 3
**Preliminary Rating:** 2
**Recommendation:** Poster
**Final Rating:** 2

**Summary:**

This paper proposes ASMR, a technique to reject updates from malicious or bad clients in Federated Learning. The method relies on outlier detection by using angular distances between gradient updates. ASMR does not require a predefined number of malicious clients during training and hence can dynamically adjust which clients to reject. Authors perform experiments under different settings --- noisy updates and unreliable clients on a histopathology dataset, showing that their method performs better or at par with other baselines.

**Strengths:**

- ASMR outperforms other baselines even when the other methods are provided with the correct number of malicious clients.
- ASMR doesn't require number of malicious clients to be hard-coded and can even dynamically adjust in case some client fails during the training

**Weaknesses:**

- This method relies on the alignment of gradient updates from different clients. The updates that do not align with the majority will be rejected. However, this can be problematic in capturing novel information from a particular client. This is one of my main concerns. How is the data distributed among different clients during the experiments? Will this method perform the same if the data is not homogenous, i.e., say, label distribution is different at different clients (or maybe the age distribution of subjects is different)?
- Similarly, the test set sizes are not clear.
- The details of the method are not clear. What is `ReachDist`? Is it the cosine distance? What is the threshold used for OF, and how is it decided?
- The claim "...persists even in scenarios where a majority of updates are malfunctioning" is not supported by experiments. It would be nice to have an experiment to support this claim.

**Detailed Comments:**

See the weakness and questions section.

Minor typo:
"we introduced noise resulting in a global model decrease ranging from 20% to 30%" -> ... Global model performance decrease...

**Justification Of Final Rating:**

I maintain my initial rating. While authors answer various questions in the rebuttal. My concern about experiments in heterogeneous data distribution settings, which is often the most interesting setting to evaluate the FL algorithm, is not addressed. Further, there are unsupported strong claims, such as being able to work with the majority of malicious clients. Hence, I am maintaining my weak reject rating.

**Justification Of The Preliminary Rating:**

This paper addresses a significant problem but lacks essential experiments. In particular, I don't know how this method would perform when clients have different data distributions, which is vital in FL settings. Hence, I would vote for weak reject.

**Questions To Address In The Rebuttal:**

- Please see weaknesses.
- What is the full form of the name "ASMR"? I didnot find what it means.
- What are small rectangle box near the top in fig 5 and 6? Do they indicate accuracy?
- How do methods that need number of malicious client peform if this value is misspecificed? It would be nice to have an experiment to show that problem.

**Special Issue:**

No

---

> ### Author Response · Authors · 2024-03-17
>
> ## Weaknesses
>
> ### This method relies on the alignment of gradient updates from different clients. The updates that do not align with the majority will be rejected. However, this can be problematic in capturing novel information from a particular client. This is one of my main concerns. How is the data distributed among different clients during the experiments? Will this method perform the same if the data is not homogenous, i.e., say, label distribution is different at different clients (or maybe the age distribution of subjects is different)?
>
> Thank you for this excelent questions.  Malfunctioning attacks maximize their performance by severely altering the updates, from this the choice of the biggest gap is that malfunctioning clients comes up naturally.. Otherwise, they are not able to corrupt the global model performance effectively, leaving the global model less vulnerable. We expect this to work for heterogeneous datasets, since the difference between a client with distribution shift is expected to be smaller than for an update that is able to corrupt the global model performance according to the analysis by [1]. The crucial part here is that one malfunctioning client alone has the potential to damage the global model performance. In our work, we assumed that malfunctioning clients are independent from each other. This means malicious clients are not able to exchange information to amplify their attacks in order to collaborate for overcoming the defense.
> [1] Li, Suyi, et al. "Learning to detect malicious clients for robust federated learning." arXiv preprint arXiv:2002.00211 (2020).
>
> ### Similarly, the test set sizes are not clear.
>
> We made a 0.7 / 0.3 train test split on the dataset. Each client receives an equally sized portion of the data that is not shared or exchanged. The server receives the test set. We revised our manuscript and added Table 2 to provide more information on this in the appendix.
>
>
> ### The details of the method are not clear. What is ReachDist? Is it the cosine distance? What is the threshold used for OF, and how is it decided?
>
> The reachability distance is a general concept using distance measures. It serves to calculate the reachability density, which is the inverse of the average reachability distance to all other updates. The reachability density is then used to determine the outlier factor as described in the paper. In our implementation, we used the cosine distance as reachability distance. To make it more clear we revised this section in the new version and just used cosine distance instead of the general reachability distance.
> We affirm that choosing the largest gap, which requires malfunctioning clients to remain significantly distant from all other updates, is crucial in mitigating their detrimental impact on the performance of the global model.Without adhering to this principle, such clients may fail to significantly affect the model's integrity. We anticipate this approach to be effective across heterogeneous datasets, as the discrepancy between a client with a distribution shift is expected to be smaller compared to an update capable of undermining the global model's performance, as indicated by the analysis in [1]. Crucially, it should be noted that a single malfunctioning client possesses the capacity to impair the overall global model's performance. Our work operates under the assumption that malfunctioning clients are independent entities, thereby precluding malicious clients from engaging in information exchange to amplify their attacks and collaborate to overcome the defensive mechanisms.
> [1] Li, Suyi, et al. "Learning to detect malicious clients for robust federated learning." arXiv preprint arXiv:2002.00211 (2020).
>
> ### The claim "...persists even in scenarios where a majority of updates are malfunctioning" is not supported by experiments. It would be nice to have an experiment to support this claim.
>
> We'd like to emphasize that our approach has the potential to work for situations where a majority of malfunctioning clients exist and do not support each other. We've made this clarification in the updated version of the paper. Thank you for the opportunity to elucidate this point.
>
> ## Detailed Comments
>
> ### Minor typo: "we introduced noise resulting in a global model decrease ranging from 20% to 30%" -> ... Global model performance decrease…
>
> Thanks for letting us know, we correct this in the new version of the paper.

---

> ### Author Response · Authors · 2024-03-17
>
> ## Questions To Address In The Rebuttal:
>
> ### What is the full form of the name "ASMR"? I didnot find what it means.
>
> The name is an acronym of the title of our paper: ASMR: **A**ngular **S**upport for **M**alfunctioning client **R**esilience. In the new version we provide an introduction of this term.
>
> ### What are small rectangle box near the top in fig 5 and 6? Do they indicate accuracy?
>
> These are box plots. Each experiment is evaluated over multiple seeds. Ten seeds for the experiment with a fixed number of malfunctioning clients and five seeds for the scenario of a dynamically changing number of clients. The boxplots present the final test performance of the trained global model. In the new version, we removed the unnecessary y-axis to scale up the plot for better visibility.
>
> ### How do methods that need number of malicious client peform if this value is misspecificed? It would be nice to have an experiment to show that problem.
>
> Great question and we happily clarify it. The consequences of undetected malfunctioning clients, including methods that require a fixed number of malicious updates, can be seen on two different plots in our paper. In the experiment, Figure 4 (page 7) & 5 (page 8) on the right side, the dynamically changing number of malfunctioning updates, the number of malfunctioning updates is set to four, while the attack probability is 75%. MKrum and DnC are set to exclude three updates in each round. This is indeed the expected number of malfunctioning clients, but it is not accurate each round. In the results it is shown that even excluding the expected number of malfunctioning clients is not accurate enough to prevent the global model from serious consequences. Further, Figure 6 shows the negative impact on the global model test accuracy, if the number of excluded clients is zero, thus the number is misspecified by three. Certainly, this is a trivial example but also here the number of clients is misspecified, illustrating that having to specify a number of malfunctioning updates can be an infeasible decision to make, as assuming there will be a set number (even 0) malfunctioning updates can be devastating for model performance.

---

> > ### Comment · Reviewer_1Cnm · 2024-03-29
> >
> > I read the response and think the experiments need to be more thorough to support the claims. In particular, the performance in heterogenous settings needs to be confirmed, however intuitive it may seem. But, I like that this approach is somewhat agnostic to a number of malicious clients.
> >
> > "Malfunctioning attacks maximize their performance by severely altering the updates." -> While I agree with this, gradients in the case of a heterogeneous setting can also be different. The smallest difference/deviation needed to corrupt updates and the maximum difference in the case of a heterogenous setting may overlap, and the authors have not shown any evidence to the contrary. For example, a sophisticated adversary may corrupt updates for only one layer and hence appear close to other updates.

---

### Official Review · Reviewer_whs3 · 2024-03-02

**Confidence:** 4
**Preliminary Rating:** 3
**Final Rating:** 3.5

**Summary:**

The authors propose a method called ASMR which aims to automatically identify malfunctioning clients in federated learning such that they do not negatively affect updates. The approach is based on calculating the angular distance between client updates and then selecting those with a larger distance as they are most likely malfunctioning. The approach is tested on a large histopathological dataset for three types of malfunctioning updates.

**Strengths:**

- The question being explored is important to the community, and the removal of the requirement to know the number of malfunctioning clients is a key improvement over existing approaches
- The approach is simple and easily applicable to different set ups
- The background work is well explained, providing strong motivation for the approach
- Three types of attacks are explored, and I appreciated the domain specific artefacts, exploring the particularities of a medical imaging application

**Weaknesses:**

- The paper is let down by its presentation and lack of clarity. Specifically, the methods section is hard to understand and the figures are all unclear and often add little. (More detailed comments below).
- There are a vast number of hyperparameters introduced, for instance the severity of attacks, the number of clients, the number of malfunctioning clients. These are not justified or explored and it is unclear what if any effect they have on the result.
- It is never made clear what the angular distance is calculated between. I assumed the gradients of client updates but it is never stated.
- The boundary of malfunctioning not malfunctioning is determined by the most substantial gap between client update distances. I am unsure how well this would generalise in the real world? A) it assumes that there must be malfunctioning clients at all times? B) the degree to which sites malfunction is presumably a continum and in real world set ups the data at local sites are non-iid and so will also have differences in updates? I'm not entirely sure that the largest gap is a gaurentee of the border between malfunctioning or not, especially for heterogenous medical imaging datasets.

**Detailed Comments:**

Introduction:
- ' there is no warranty about the utility of these updates' - I'd reword this for clarity, its not the standard use of warranty
- Figure 1 and 2 could be combined. Figure 1 is not adding much information.

Methodology:
- This section needs reworking for clarity. Lots of important information is in the wall of text in malfunctioning updates - maybe a table with values or something would help break it up
- SFA and ANA are used before introduction
- How were all the values chosen? how sensitive are the results to these choices?
- state what you calculate the angles between
- explain reachability distance

Experiments:
- The data section is lacking information, explain client numbers, data splits between clients etc here.
- What aggregation scheme are you using?
- There are no details for how local training is completed? All of these are needed for the paper to be replicable.
- Figure 4: what does this show? The caption states it underscores the impact of ten clients but it is unclear how. Adding error bars would also improve this figure.

Results:
- The graphs/boxplots are too small to read, especially the box plots. What does the changing colour indicate?
- Details about experimental set up eg number of malfunctioning clients should probably be in the previous section
- plotting the distances wrt to the threshold chosen would help with understanding the dynamic choice.

**Justification Of Final Rating:**

The paper presents a simple method to solve a problem of interest to the community. I tend towards accept because of the strengths compared to existing methods but am concerned about the evaluation on only homogenous data which is unrealistic in medical settings and likely to effect the choice of the decision boundary.

**Justification Of The Preliminary Rating:**

- The approach shows promise and explores an important question but is let down by presentation. My major concern is around the choice of the decision boundary, and the justification of this in the rebuttal is key to whether the paper should be accepted or not.

**Questions To Address In The Rebuttal:**

- My major concern is around the boundary. This needs to be justified, especially for application to heterogenous data at sites, as the data used here was from a single dataset. When will the choice of the largest difference be appropriate and not? Is there any mathematical justification for the choice?

---

> ### Author Response · Authors · 2024-03-17
>
> ### There are a vast number of hyperparameters introduced, for instance the severity of attacks, the number of clients, the number of malfunctioning clients. These are not justified or explored and it is unclear what if any effect they have on the result.
>
> Our objective was to illustrate federated learning scenarios encompassing various types of malfunctioning client setups. To achieve this, we employed three distinct approaches: Additive Noise Attack (ANA), Sign Flipping Attack (SFA), and pathology-specific artifacts. Each approach showcases a different form of malfunctioning, thereby representing varying degrees of severity in their impact on the global model.
> To simulate a cross-silo scenario (e.g. collaborating hospitals) we chose a number of ten clients. The amount of malfunctioning clients was chosen to be a fair number on the one hand, but remaining a serious threat on the other hand. To do so we evaluated different scenarios. In the first, a fixed number of three clients (30%) is considered. This is not a real world scenario but aligns with the assumptions of methods like MKrum and DnC. The second scenario is a dynamically changing number of malicious clients, that is more aligned to real world scenarios. This emphasizes the need for methods that can dynamically adapt to the number of malfunctioning clients. We chose a number of four malfunctioning clients (40%), but each of them is sending a malfunctioning update by a probability of 75%. This results in an expected number of malfunctioning updates of three each round. Methods that exclude a fixed number of malfunctioning updates each round, have still an educated guess, but our experiments aim to demonstrate that this is still not enough to protect the global model performance. The severity of ANA, SFA and artifacts were chosen to be:
> - SFA: A heavy attack, were even a single included update can lead to random performance of the global model. To achieve these updates a very far away from the benign ones, to corrupt the FedAvg algorithm. For this reason SFAs are easier to detect, but demonstrate the need for a dynamically changing decision boundary well.
>
> - ANA: In general the severity of ANAs are adaptable to the amount of noise that is added to the parameters before sending them to the server. For our purposes we chose an amount that damages the global model by around 20% loss of accuracy. In comparison to the SFAs the local model parameters at ANAs are way closer to the benign parameters, which makes them more difficult to detect. This task serves to demonstrate the inherent detection capabilities of each method.
>
> - Artifacts: This is another manipulation of the local model parameters. The local model parameters are not manipulated after the local training, but the training data is corrupted. The local models learn wrong features, which leads to another structure of malfunctioning local model parameters. This is aligned with real world scenarios in histopathology, and serves to prove that the detection methods are also robust against this structure of manipulation. Furthermore we wanted the artifacts to cause serious damage, but not lead to random global model accuracy. Therefore we scaled the artifacts such that they lead to a performance decrease of around 50% such that the severity is between ANAs and SFAs.
>
> We covered different malfunctioning scenarios with different structures in the updates and different severities. Finally, we conducted an experiment to show that the ASMR is robust against all those scenarios while dynamically adapting to the number of malfunctioning updates each round.
>
>
> ### Figure 1 and 2 could be combined. Figure 1 is not add much information
> Thank you for this suggestion, we thought about combining both figures, but decided to edit Figure 2, to extend the information contained in it by a visualization of how clients are selected for the aggregation and how clients are excluded. Figure 1 serves as a visualization of the problem for people who are less familiar with federated learning. It is crucial for us to clarify the motivation behind our work, including a clear understanding of what we mean by malfunctioning - unreliable or malicious - clients. We aim to emphasize that malfunctioning does not necessarily imply malicious intent. Unreliable clients are an important aspect in the application of federated learning in histopathology, since low quality data is a persistent challenge. This is a previously unexplored aspect and therefore we think it is important that Figure 1 draws attention to this characterization, so we decided to keep it in the new version of the paper.

---

> ### Author Response · Authors · 2024-03-17
>
> ### It is never made clear what the angular distance is calculated between. I assumed the gradients of client updates but it is never stated.
>
> In each round, the clients transmit their local model parameters to the server. Subsequently, the pairwise cosine distance between all updates is calculated providing insights into the distance between each update and every other update.
> Indeed, this part was not described in this paper. We added this information to the new version.
>
>
> ### The boundary of malfunctioning or not malfunctioning is determined by the most substantial gap between client update distances. I am unsure how well this would generalise in the real world? A) it assumes that there must be malfunctioning clients at all times? B) the degree to which sites malfunction is presumably a continum and in real world set ups the data at local sites are non-iid and so will also have differences in updates? I'm not entirely sure that the largest gap is a gaurentee of the border between malfunctioning or not, especially for heterogenous medical imaging datasets.
>
> - A: Yes, like all other methods the presence of malfunctioning clients is expected. The absence of malfunctioning clients would lead to the exclusion of benign clients. On the other side, each round would still contain only benign clients, such that the global model wouldn't suffer.
>
> - B: To corrupt the accuracy of the global model performance the, the malfunctioning updates need to be far off from all other updates. Our interpretation of the malfunctioning client setup is that updates are denoted as malfunctioning when they are so far away from the regular ones that even a single malfunctioning update is capable of damaging the global model performance regardless of the intention behind it. On the other hand, an update that is not that far away from the benign ones, even if sent with malicious intentions, would be “averaged out” and does not need to be defended. The idea to search for the biggest gap is inspired by an analysis of [1], where they argue that distribution shifts are less heavy than malfunctioning updates.
>
> [1] Li, Suyi, et al. "Learning to detect malicious clients for robust federated learning." arXiv preprint arXiv:2002.00211 (2020).
>
> ### ' there is no warranty about the utility of these updates' - I'd reword this for clarity, its not the standard use of warranty
> We totally agree with this. In the new version of our paper, we changed it from “warranty” to “guarantee”, since we think that this is more aligned to the core of this sentence. Thank you for this suggestion, we appreciate the effort very much to clarify the paper and make it more easily readable.
>
> ## Methodology
>
> ### This section needs reworking for clarity. Lots of important information is in the wall of text in malfunctioning updates - maybe a table with values or something would help break it up
>
> Thanks for the suggestion, we add  table 1 and 2 containing important information to the appendix. This table contains the number of clients, the number of malfunctioning clients, the attack probability, the aggregation method, the local epochs and more.
>
>
> ### SFA and ANA are used before introduction
>
> SFA and ANA were introduced in the introduction of this paper. We agree that introducing ANA and SFA in the methodology section again might be confusing. We edit this in the new version of our paper to be only introduced once. Thanks for letting us know.
>
>
> ### explain reachability distance
>
> The reachability distance is a general concept using distance measures. It serves to calculate the reachability density, which is the inverse of the average reachability distance to all other updates. Subsequently the reachability density is used to determine the outlier factor as described in the paper. In our implementation, we used the cosine distance as reachability distance. To enhance clarity, we have revised this section in the updated version.

---

> ### Author Response · Authors · 2024-03-17
>
> ## Experiments
>
> ### The data section is lacking information, explain client numbers, data splits between clients etc here.
>
> We consider the case of ten clients in total in our experiments. The number of malfunctioning clients is either three (100% probability of sending malfunctioning updates) or four (75% probability). This is described in the results section. For clarification purposes, we moved it into the experiment section. We also add the information about the data split in the new version of our paper in Table 2 in Appendix A.
>
> ### What aggregation scheme are you using?
> The clients send their local model parameters to the global server. Then the updates are aggregated with FedAvg. We chose FedAvg as it is most common baseline for aggregation in FL. We added this information to the new version of the paper.
>
>
> ### There are no details for how local training is completed? All of these are needed for the paper to be replicable.
>
> This is answered above, please see previous points.
>
> ### Figure 4: what does this show? The caption states it underscores the impact of ten clients but it is unclear how. Adding error bars would also improve this figure.
>
> Figure 4 (now Figure 6 in the Appendix) shows the impact of the absence of defense methods. These curves show the global model test performance of a system of ten clients, where three of them are malfunctioning. As we consider the three malfunctioning schemes ANA, SFA and unreliable clients through pathology specific artifacts we wanted to demonstrate the serious consequences, when those malfunctioning clients are not excluded from aggregation. This also demonstrates our choice of the severity of those malfunctioning as described above.
>
> ## Results
>
> ### The graphs/boxplots are too small to read, especially the box plots. What does the changing color indicate?
> Thank you for this valuable advice and we agree to it. We removed the unnecessary y-axis labels to remove space between the plots and make them more dense. Two color schemes are used. One for the experiment with a fixed number of malfunctioning clients and the other for the experiment with a dynamically changing number of clients. The color gradient within the color schemes is to differentiate the different methods easier.
>
> ### Details about experimental set up eg number of malfunctioning clients should probably be in the previous section
>
> Thanks for the advice. We put the description of our experimental setup in the Experiments section. Additionally we add a table containing the details to Appendix A. We hope you find this more enjoyable to read.
>
> ### plotting the distances wrt to the threshold chosen would help with understanding the dynamic choice.
> In the updated version of the paper, we revised Figure 3 to illustrate the concept of dynamic selection. Beneath the visualization depicting client support, we incorporated a representation of the outlier factor. Using a color scale ranging from green to red, the level of the outlier factor is depicted. Notably, benign clients exhibit significantly lower outlier factors compared to malicious ones. Thus, we visualized the decision boundary at the largest gap.
>
> ## Questions To Address In The Rebuttal
>
> ### My major concern is around the boundary. This needs to be justified, especially for application to heterogenous data at sites, as the data used here was from a single dataset. When will the choice of the largest difference be appropriate and not? Is there any mathematical justification for the choice?
>
> We argue the choice of the biggest gap is that malfunctioning clients need to be far away from all other updates. Otherwise, they are not able to corrupt the global model performance. We expect this to work for heterogenous datasets, since the difference between a client with distribution shift is expected to be smaller than for an update that is able to corrupt the global model performance. The crucial part here is that one malfunctioning client alone has the potential to damage the global model performance. In our work, we assumed that malfunctioning clients are independent from each other. This means malicious clients are not able to exchange information to amplify their attacks in order to collaborate for overcoming the defense. From this it is unlikely that scenarios that optimize the malicious clients would be much harder to detect, however, it is also a common assumption made in previous works. In conclusion, we agree that this kind of setting should be further explored in future works as coordinated attacks present a harder challenge not yet explored. However, we wanted to first show that our method outperforms other methods in this already easier and more common setting, hoping it will be a stepping stone for upcoming research.

---

> > ### Comment · Reviewer_whs3 · 2024-03-20
> > **Response to authors**
> >
> > The authors have done a good job of addressing comments and updating their manuscript. Overall I think the paper is of interest the community.
> >
> > Remaining comments:
> > - *_'In conclusion, we agree that this kind of setting should be further explored in future works as coordinated attacks present a harder challenge not yet explored. However, we wanted to first show that our method outperforms other methods in this already easier and more common setting, hoping it will be a stepping stone for upcoming research.'_*
> >
> > I think the limitation of your experiments being conducted on homogenous data and therefore not knowing the effect of heterogenous data on the decision boundary needs to be discussed in the manuscript.
> >
> > - Fig 3 is too big

---

> > > ### Author Response · Authors · 2024-03-27
> > >
> > > Thanks for the valuable feedback. We revised Figure 3 to achieve a denser representation. Additionally, we addressed the limitations of our experiments in the Discussion section.

---

### Meta-Review · Area_Chair_AENY · 2024-04-03

**Recommendation:** Accept (Poster)
**Confidence:** 4

**Metareview:**

The Meta-Reviewer has read all reviews and rebuttal of authors.

Reviewers have all confirmed they have taken the author rebuttal into account for their final recommendation.

The work has received mixed reviews.
The primary original concerns of the reviewers seemed to be about clarity/presentation/writing of the paper, limitations of the evaluation, and the big question (and potential limitation) of whether the method would work in the setting when different clients have heterogeneous data. During the rebuttal, the authors have improved clarity/writing of the paper, acknowledged by reviewers. The rebuttal discussion also provided explanations about the evaluation. These improvements were acknowledged by reviewers that have increased their final scores. It remains unclear whether the method would be beneficial in practical settings in the case of heterogeneous data.

Taking all points into account, there seems to be a concensus that the work is of adequate quality, and provides insights into the problem along with exploration of a method that has advantages over existing works, which could be of interest to the community. Therefore I will recommend acceptance of the work.

If the work is accepted, I hope the authors would continue improving the paper until a camera-ready, for example by incorporating into the paper the clarifications made during the rebuttal (eg about comparisons with previous work, stating explicitly the limitation about homogeneous data and how the method could interact with heterogeneous data, etc).

---

### Decision · Program_Chairs · 2024-04-05

Accept (Poster)